# The Role of Intracellular Lipid-Binding Proteins in Digestive System Neoplasms

**DOI:** 10.3390/curroncol32100531

**Published:** 2025-09-24

**Authors:** Christos Kakouratos, Adriana Fernandez Garcia, Pramod Darvin, Hemant M. Kocher

**Affiliations:** 1Centre for Tumour Biology, Barts Cancer Institute—A CRUK Centre of Excellence, Queen Mary University of London, London EC1M 6BQ, UK; c.kakouratos@qmul.ac.uk (C.K.); a.fernandezgarcia@smd23.qmul.ac.uk (A.F.G.); p.darvin@qmul.ac.uk (P.D.); 2Barts and the London HPB Centre, The Royal London Hospital, Barts Health NHS Trust, London E1 1FR, UK

**Keywords:** lipid chaperones, transport, fatty acid binding protein, cellular retinoic acid binding protein, gastrointestinal cancers, intracellular lipid binding proteins (iLBPs)

## Abstract

Fats and fat-soluble vitamins, such as retinoic acid, are insoluble in water, so they need help for their transportation within the cells. Intracellular lipid-binding proteins help the cells to absorb and use these types of molecules for energy, growth, and repair. In gastrointestinal cancers, these proteins can influence how quickly tumours grow, or how far they spread, or how they respond to treatment. Some of these proteins may help cancer cells resist drugs or survive, while others may slow cancer progression. Understanding how these proteins work can help clinicians and scientists predict cancer behaviour and develop new treatments, offering more effective and personalised care for patients.

## 1. Introduction

### 1.1. Intracellular Lipid-Binding Proteins and Their Function

Intracellular lipid-binding protein (iLBPs) are a family of ubiquitous, low molecular mass proteins facilitating solubilisation and transport of essential cellular lipids, fatty acids (FAs), and other hydrophobic molecules [1,2,3,4], whose genes are dispersed across several different chromosomes, classifying iLBPs as a multigene family [4,5,6]. It has been postulated that this family originated from a single ancestral gene through a minimum of fourteen gene duplications events [7,8], with a conserved gene structure including four exons separated by three introns. Furthermore, the relatively short length (16–60 amino acids) of the exons has been preserved, resulting in the positions of the introns within the genes remaining essentially identical [4,5,6]. Interestingly, chromosome 8 in humans harbours five of these loci (FABP4, FABP5, FABP8, FABP9, and FABP12). Despite organisms exhibiting a considerable variation in amino acid identity (20–70%), crystal structures of proteins demonstrate a consistent overall tertiary structure. This characteristic structural arrangement features a cavity formed by ten anti-parallel strands and two alpha helices, which can bind and hold lipophilic compounds such as fatty acids [1,5,6]. Nevertheless, iLBPs are classified into four subfamilies based on their ligand-binding preferences [1].

Although the complete physiological functions of this diverse family need further research, they seem to have a role as carriers, shuttling ligands through the cytosol and facilitating the efficient uptake of endogenous lipids into the tissues, thus modulating rates of ligand access, availability, utilisation, and metabolism [2,9]. Moreover, fatty acid binding proteins (FABPs), the most prominent of iLBPs, are expressed in major organs involved in xenobiotics absorption, distribution, and metabolism, making them potential targets for drug development [4]. Specifically, the importance of these proteins lies on the intestinal absorption of lipids and the utilisation of free fatty acids from plasma by key organs, such as liver, muscle, amongst other tissues, by helping in the translocation of FAs from the cell surface to the endoplasmic reticulum and mitochondria, thus underlining their vital role in the absorption and cellular utilisation of FAs, as well as certain lipid-soluble drugs and toxins [10]. Increasingly, it is recognised that FABPs may also contribute to regulate oxidative and inflammatory states, which is particularly relevant in maintaining nervous system homeostasis [11].

### 1.2. Structure

The protein structure primarily consists of a β-sheet conformation, making up 70% of its composition. This β-sheet is composed of ten anti-parallel β-strands that arrange into two perpendicular ß-sheets, each bonded by hydrogen bonds and folded to adopt the shape of a barrel (Figure 1; PDB ID: 2FR3 [12,13,14]). Positioned between the first and the second ß-strand are two alpha-helices arranged in a helix–turn–helix motif loop, a key feature for flexibility, enabling it to accommodate around larger ligands by opening and closing around them, and this feature is facilitated by the gap between βD and βE strands [4,5,15].

The ligand binding site, known as the ligand portal region, is composed of the helix–turn–helix motif and loops between βC-to-βD and βE-to-βF [16]. This region occupies a space of 400–600 A, which represents 5–10% of the protein’s total volume and it is two to three times larger than needed to hold its ligand. This region serves as a ligand portal because structural exposure, mobility, protease sensitivity, and the impact of mutations in this area affects binding kinetics. Within the ligand-binding cavity, there are ten conserved water molecules that form part of the ligand contact side. Ligands are stabilised within the amino acid sidechain that line the binding cavity by ionic interactions, hydrogen bonding networks with water, and interactions with hydrophobic regions. These findings have been shown by crystallographic and nuclear magnetic resonance (NMR) studies of iLBPs [4,5,15]. Figure 1 depicts the binding of the ligand, all-trans-retinoic acid (atRA) in the binding site of cellular retinoic acid binding protein 2 (CRABP2).

### 1.3. Subtypes, Tissue Distribution, Expression, and Ligand Binding

The heterogeneity of iLBPs is represented by four different subfamilies, categorised based on their ligand-binding preferences, including cellular retinol (CRBP), cellular retinoic acid (CRABP), and fatty acid (FABP) binding proteins. CRBPs and CRABPs belong to subfamily I, while the remaining members, FABPs, are grouped into three different families. Subfamily II comprises FABP1 and FABP6, subfamily III includes only FABP2, and subfamily IV, which includes most of the members, encompasses FABP3-5, FABP7-9, and FABP12 (Figure 2) (Table 1) [1,17].

There are four distinct CRBPs. Among them, the structure of CRBP1 has been extensively investigated by X-ray crystallography, NMR, and proteolytic digestion, followed by mass spectrometric analysis of the fragments [18]. These studies have characterised the structure of CRBP1, with CRBP1 and CRPB2 sharing 56% primary amino acid identity. The major differences between CRBP1 and CRBP2 are localised to the alpha2 helix, which account for their different interactions with enzymes [3]. CRBPs appear to be more specific for retinol and retinal ligands, as well as monoacylglycerols [19]. While all four subtypes bind all-trans retinol, CRBP1, CRBP2, and CRBP4 also binds 13-cis and 9-cis retinol [20].

CRABPs share 78% identity, with variations in the ligand access portal that are crucial for retinoic acid receptor (RAR) binding [3]. The distinct expression patterns of CRABP1 and CRABP2 suggest they accommodate different ligand requirements, mostly retinoic acid (RA), in various tissues [21]. Generally, CRABPs serve to solubilize and protect their ligand in the cytoplasm and transport RA between different sub-cellular compartments, such as the nucleus [22]. CRABP1 and CRABP2 bind to retinol (vitamin A) and all-trans-retinoic acid (atRA) or RA with different affinities (Figure 2) [3]. Specifically, atRA appears to have a higher binding affinity for CRABP1 than for CRABP2 [23]. CRABP1 moderates cellular response to RA by sequestering it and/or facilitating its catabolism [24]. In contrast, CRABP2 resides in the cytoplasm, specifically binds to atRA, and enhances its action by delivering it to receptors in the nuclear compartment [25].

The remaining three subfamilies are composed of FABPs members, which display unique tissue-specific expression patterns. Initially, FABPs were named after the tissue where they were discovered [5,26]. Each isoform contains five conserved amino acid sites, with variations in amino acids that form the cavity ligand-binding site led to each FABPs distinct capacity to bind various ligands [27].

FABP1 and FABP6 (Subfamily II) have larger binding pockets, allowing them to accommodate larger ligands, such as bile acids, cholesterol, bilirubin and heme, and thus, their role is predominantly in the gastro-intestinal tract and liver. Subfamily III consists solely of FABP2 that has a smaller, solvent accessible binding pocket, giving it a higher affinity for saturated long-chain fatty acids (LCFAs) [2,4]. The members of subfamily IV, which is the biggest among the iLBPs subfamilies, bind diverse lipids, as illustrated in Figure 2.

FABPs perform various functions, such as transporting FAs from the plasma membrane into the cytoplasm, thereby accelerating FA uptake [9], with an important role regulating signalling pathways by delivering lipids to specific transcription factors and nuclear receptors, such as the peroxisome proliferator-activated receptor (PPAR) family [6] and the retinoic acid receptor (RAR) and the retinoid X receptor (RXR) [28].

### 1.4. Ligands of iLPBs

Fatty acids (FAs), straight alkyl chains ending with a carboxyl group, are essential components of cellular membranes, form complex lipids (glycerolipids, phospholipids, and sphingolipids), and act as signalling molecules triggering activation of specific transcription factors [6], and these require transport to key sites of action within the cell in a predominantly water-soluble cytoplasm. They exist in various forms including saturated, monounsaturated, and polyunsaturated fatty acids (PUFAs). Saturated fatty acids (animal fats, some vegetable oils) have no double bonds, while monounsaturated fatty acids (MUFAs), like oleic acid, and PUFAs docosahexaenoic acid (DHA) and linoleic acid, contain one or more double bonds [29,30]. For example, DHA is key component of phospholipids of neuronal and synaptic membranes making it crucial for the growth, survival, and maintenance of neurons as well as functioning as a neurotrophic factor, regulating synaptic activity, and is involved in anti-inflammatory signalling [31]. Thus, iLPBs are key to the movement of fatty acids to specific cellular locations, influencing processes such as gene expression, cell growth, and differentiation [32].

Retinoids represent another significant group of ligands, over 4000 different derivatives of vitamin A have been shown to play a crucial role in essential cellular processes of the body such as vision, immune function, reproduction, differentiation, and maintenance of epithelial tissues [33]. This group encompasses retinol (vitamin A) and its major metabolites: retinal and retinoic acid (RA) or atRA [3,24]. RA plays a pivotal sub-cellular function of pattern formation, growth, and differentiation [21,24,34]. For example, during differentiation, RA alters expression of genes which function as transcription factors, RA metabolism and transport proteins, protooncogenes, apoptosis-related proteins, and growth factors, among others [33]. Thus, RA shows promise as a therapeutic anti-cancer option and is used in treating human promyelocytic leukaemia, and in the laboratory, shown to affect melanoma, neuroblastoma, and mouse teratocarcinoma stem cells [24,34,35,36].

Furthermore, retinoids can act as both chemo-preventative and chemotherapeutic agents, potentially preventing or reversing cell tumourigenesis [33]. For instance, several studies have demonstrated the chemo-preventive effects of retinoids in breast cancer [37], renal-cell carcinoma [38], and other precancerous lesions [33]. Key mechanisms of transport of retinoids, such as those with iLPBs may thus influence its anti-cancer properties, particularly in gastro-intestinal cancers, and each molecule will be discussed in detail.

## 2. The Role of Intracellular Lipid Binding Proteins in Gastrointestinal Diseases

### 2.1. CRBPs

CRBPs exhibit a high-affinity binding for retinol, functioning as chaperones to control the prenuclear phase of retinoid signalling [39,40]. CRBPs likely sequester most cellular retinoid in the cytoplasm due to CRBP’s higher concentration compared to their ligands, and their high affinity for them [3]. Since, retinoids can inhibit the formation of preneoplastic lesions and prevent the appearance of metastasis [40,41], CRBP may play an oncogenic role in this regard. On the other hand, CRBPs also regulate retinoid esterification and storage in liver [42], protecting retinol from being metabolised by enzymes [43], and thus may have an anti-cancer effect.

Different CRBPs have been studied based on tissue-specific expression: CRBP1 is widely expressed in epithelial tissues and is best studied [44,45] (Table 2), whereas CRBP2-4 has more restricted tissue distributions and is also poorly studied [20].

In several tumour types, particularly the more aggressive ones, CRBP1 expression levels are reduced [44,45]. In pancreatic cancer, the expression of CRBP1 is reduced in a substantial proportion of patients, with complete loss observed in approximately 34% of cases. Despite this frequent downregulation, there is no significant association between CRBP1 expression levels and patient outcome [46]. In pancreatic cancer, CRBP1 loss alone is insufficient to induce carcinogenesis or alter sensitivity to retinoid-based therapy [47].

Similar results have been described for CRBP1 in oesophageal squamous cell carcinoma (ESCC), as its promoter methylation was observed exclusively in advanced-stage tumours and was significantly associated with reduced mRNA expression, suggesting that epigenetic silencing of CRBP1 may contribute to tumour progression in ESCC. This promoter methylation, which is sometimes seen even in premalignant lesions, often coincides with retinoic acid receptor beta 2 (RARβ2) hypermethylation [40]. This epigenetic silencing has been observed in advanced stages in various tumours including oesophageal, gastric, and pancreatic cancer [47,48,49,50]. CRBP1 is downregulated in hepatocellular carcinoma (HCC), with higher expression linked to better overall survival (OS) [51,52].

In HCC, CRBP1 has been shown to alter cancer cell stemness by suppressing Wnt/ß-catenin signalling pathway. Specifically, CRBP1 increases intracellular levels of RA, inducing the activation of RAR/RXRs which lead to the transcriptional expression of a secreted antagonist of the Wnt/ß-catenin signalling pathway, the Wnt inhibitory factor 1 (WIF1) [52].

Interestingly, CRBP1 expression can act as a radiation-sensitivity predictor in rectal cancer, where the promoter hypermethylation significantly correlates with histological response to neoadjuvant radiotherapy. Higher CRBP1 expression levels tend to indicate radiation sensitivity, whereas lower levels suggest resistance [53].

**Table 2 curroncol-32-00531-t002:** Role of CRBP1 in GI cancers.

Cancer Type	Role of CRBP1	Details	Reference
Pancreatic Cancer	Tumour suppressor (likely)	Frequently downregulated; no association with patient outcome; CRBP1 loss alone does not induce carcinogenesis or impact retinoid therapy sensitivity.	[46,47]
Oesophageal Squamous Cell Carcinoma (ESCC)	Tumour suppressor	Promoter methylation seen only in advanced stages; correlates with reduced mRNA expression; often occurs with RARβ2 methylation.	[40,48]
Gastric Cancer	Tumour suppressor	CRBP1 promoter methylation observed in advanced stages.	[50]
Hepatocellular Carcinoma (HCC)	Tumour suppressor	Downregulated in tumours; higher expression linked to better OS.	[51,52]
Rectal Cancer	Biomarker for radiotherapy sensitivity	Promoter hypermethylation correlates with poor histological response to neoadjuvant radiotherapy; high expression indicates sensitivity.	[53]

### 2.2. CRABP1

Cellular retinoic acid-binding protein 1 (CRABP1) is a cytosolic carrier that binds all-trans retinoic acid (atRA) with high affinity and regulates its intracellular trafficking. By selectively delivering atRA to cytochrome P450 enzymes (CYP26), CRABP1 facilitates the metabolic degradation of retinoic acid, thereby modulating its availability for nuclear receptor signalling and maintaining retinoid homeostasis [21]. Thus, CRABP1 is crucial for retinoic acid-mediated differentiation and proliferation processes [54]. CRABP1 expression appears to be reduced in many the cancer types, suggesting a tumour suppressor role, although it acts as tumour promoter in others (see Table 3).

Similar to CRBPs, CRABP1 has been implicated in tumour suppression, with studies showing that its mRNA levels are reduced in certain human epithelial tumours (colorectal and hepatocellular cancers) due to hypermethylation of its promoter region (Table 3) [55,56]. Conversely, elevated levels of CRABP1 expression have been reported in pancreatic neuroendocrine tumours (pNETs) alongside association with lymph nodes metastasis, poor differentiation, and may serve as a prognostic biomarker for pNETs [57].

**Table 3 curroncol-32-00531-t003:** Role of CRABP1 in GI cancers.

Cancer Type	Role of CRABP1	Details	Reference
Colorectal Cancer	Tumour suppressor	mRNA levels are reduced due to hypermethylation of the promoter region.	[56]
Hepatocellular Carcinoma (HCC)	Tumour suppressor	Reduced expression due to promoter hypermethylation.	[55]
Pancreatic Neuroendocrine Tumours (pNETs)	Tumour promoter/Prognostic biomarker	Elevated expression associated with lymph node metastasis, poor differentiation; may serve as a prognostic biomarker.	[57]

### 2.3. CRABP2

CRABP2 is also a cytosolic chaperone that binds atRA with high affinity and facilitates its cellular trafficking [21,58]. However, whilst CRABP1 directs atRA toward catabolism, CRABP2 enhances its signalling by acting as a nuclear shuttle for atRA to interact with nuclear receptors (RAR/RXR), thereby facilitating transcriptional activation and affecting cell proliferation, differentiation, and apoptosis, though the exact mechanism for this differential function between CRABP1 and 2 is poorly understood [21,58,59,60].

CRABP2 demonstrates variable roles in gastrointestinal (GI) cancers, functioning either as a tumour suppressor or as a promoter depending on cancer type and molecular context. In oesophageal squamous cell carcinoma (ESCC), CRABP2 expression is significantly downregulated in tumour tissues relative to normal epithelium with the reduction correlating with advanced TNM stage, poor differentiation, tumour infiltration, and worse overall survival. Overexpression of CRABP2 in EC109 cells resulted in decreased proliferation and increased apoptosis and decreased cancer cell migration [61].

In contrast, CRABP2 is markedly upregulated in oxaliplatin-resistant tumours and functions as a mediator of chemoresistance [62]. It promotes resistance by binding BCL2 associated X (BAX) and parkin RBR E3 ubiquitin protein ligase (PARKIN), facilitating BAX ubiquitination and degradation, thereby inhibiting mitochondrial apoptosis. CRABP2 expression is driven by ten-eleven translocation 1 (TET1)-mediated DNA hydroxymethylation of its promoter, and TET1 knockdown reverses this upregulation and restores drug sensitivity. Using patient-derived xenograft (PDX) models, the authors showed that CRABP2 silencing enhances the anti-tumour effects of oxaliplatin. These findings identify CRABP2 as a promising target for overcoming oxaliplatin resistance in gastric cancer [62].

In pancreatic ductal adenocarcinoma (PDAC), CRABP2 emerges as a controversial but perhaps a key player. On one hand, CRABP2 functions as a key determinant of sensitivity to atRA, with its expression inversely correlated to FABP5 expression (Figure 3). PDAC cell lines characterised by high CRABP2 and absent FABP5 expression are highly responsive to atRA, showing increased apoptosis, cell cycle arrest, morphological differentiation, and reduced migration and invasion. In contrast, FABP5^high^ CRABP2^null^ cells are resistant and exhibit enhanced migration potential upon atRA exposure. Ectopic expression of CRABP2 in resistant lines resensitises them to atRA by redirecting retinoic acid signalling away from the FABP5–PPARβ/δ pathway, restoring apoptotic and anti-invasive responses. CRABP2 expression is frequently silenced in PDAC via promoter methylation but can be reactivated by demethylating agents [63].

In contrast, other studies have shown that CRABP2 is consistently overexpressed in human PDACs, with no detectable expression in normal pancreatic tissue or chronic pancreatitis [64,65]. Its expression progressively increases from low-grade PanIN lesions to invasive PDAC and is strongly positive in all primary and metastatic tumours. High CRABP2 expression is also associated with poor tumour differentiation, increased recurrence, and significantly reduced overall survival [64,65]. These findings suggest a role for CRABP2 in PDAC development. Silencing or knockout of CRABP2 significantly impairs cell migration and invasion without affecting proliferation by enhancing the stability of IL-8 mRNA through direct interaction with the RNA-binding protein human antigen R (HuR). This results to IL-8 upregulation and its downstream effectors matrix metallopeptidase 2 and 14 (MMP-2, MMP-14), key mediators of extracellular matrix remodelling [66]. CRABP2 has been found to also stabilize the Sterol regulatory element-binding transcription factor 1 (SREBP-1c) mRNA via HuR binding in an RA-independent manner, thereby promoting lipid raft cholesterol accumulation and sustaining AKT survival signalling mediating gemcitabine resistance. Knockout of CRABP2 restores gemcitabine sensitivity and enhances apoptosis by reversing the upregulation of genes involved in the synthesis of cholesterol such as SREBP-1c, 3-hydroxy-3-methylglutaryl-CoA reductase (HMGCR), and low-density lipoprotein receptor (LDLR). Notably, pharmacological targeting of CRABP2 using SNIPER-11 reduces cholesterol content, suppresses AKT activation, and sensitises tumours to gemcitabine both in vitro and in patient-derived xenograft models [65]. Thus, the role of CRABP2 in PDAC is context dependent.

In HCC, CRABP2 seems to function as tumour promoter as it is significantly upregulated in tumour tissues and cell lines, correlating with disease progression [67]. Silencing CRABP2 in HCC cell lines suppresses proliferation, migration, invasion, and colony formation, while promoting apoptosis in vitro [67] with inhibition of the ERK–VEGF signalling pathway and upregulates pro-apoptotic markers such as Bax and cleaved caspase-3. In vivo, CRABP2 knockdown reduces tumour growth and angiogenesis, indicating that CRABP2 promotes HCC progression through survival and pro-metastatic signalling [67].

In colorectal cancer (CRC), CRABP2 is significantly upregulated in tumour tissues compared to normal epithelium and its role appears to be even more complicated [68,69]. CRABP2 displays a dual and compartmentalised role, influencing both tumour growth and metastasis through distinct nuclear and cytoplasmic mechanisms [68]. On one hand, nuclear CRABP2 promotes tumour growth by enhancing proliferation and suppressing apoptosis through direct interaction with the tumour suppressor retinoblastoma protein (RB1). CRABP2 binding leads to RB1 destabilisation, which facilitates E2F-mediated transcription and cell-cycle progression. In mouse models and CRC cell lines, CRABP2 knockout markedly reduced tumour burden, increased apoptosis, and suppressed Ki-67 expression, highlighting its tumour-promoting function. On the other hand, cytoplasmic CRABP2 exhibits metastasis-suppressive activity in the liver by interacting with the mitochondrial protease ATPase family gene 3-like 2 (AFG3L2). This interaction stabilises the PTEN-induced kinase 1 (PINK1) and activates mitophagy, which mitigates mitochondrial dysfunction and reduces reactive oxygen species accumulation. Through this pathway, CRABP2 inhibits liver metastasis in vivo. However, this anti-metastatic effect is modulated through compartment-specific mechanisms, with CRABP2 promoting proliferation when it is localised in the nucleus and suppressing metastasis when it is localised in the cytoplasm via mitochondrial regulation [68].

In another interesting study, Hao et al. demonstrated that CRABP2 also facilitates peritoneal metastasis (PM) in CRC by promoting the epithelial-to-mesenchymal transition (EMT) and invadopodia formation [69]. High CRABP2 expression is associated with advanced tumour stage, reduced overall survival, and enriched expression in peritoneal metastatic lesions. According to the study, CRABP2 activates the Transforming Growth Factor beta (TGF-β)/Smad pathway, increasing EMT markers, such as Snail, β-catenin, and Vimentin, while suppressing epithelial markers like ZO-1 and E-cadherin. This promotes invasive behaviour and supports tumour–mesothelial adhesion via CDH1-mediated cell–cell communication. Furthermore, CRABP2 upregulates MMP14 and reorganises the actin cytoskeleton to drive invadopodia formation, facilitating ECM degradation. High CRABP2 expression also correlates with immunosuppressive tumour microenvironments, marked by low CD8^+^ and CD4^+^ T cell infiltration and enrichment of M2 macrophages [69]. The tumour promoting and tumour suppressor roles for CRABP2 are summarised in Figure 4 and Table 4.

Differences between CRABP1 and CRABP2 expression and function have been explored extensively, but their exact roles remain unclear. They exhibit distinct functional differences in terms of RA signalling. CRABP1 dissociates RA to facilitate its interaction with RAR, whereas CRABP2 directly interacts with RAR during the movement of RA [21]. Furthermore, CRABP1 and CRABP2 play opposite roles depending on the RA sensitivity of cells. In RA-sensitive cells, CRABP1 stimulates proliferation and resistance to RA, whereas CRABP2 reduces both. Conversely, in more RA-resistant cells, their roles are reversed [36]. Understanding the involvement of RA resistance and CRABPs is crucial for effectively utilizing RA in cancer therapy.

**Table 4 curroncol-32-00531-t004:** Role of CRABP2 in GI cancers.

Cancer Type	Role of CRABP2	Details	Reference
Oesophageal Squamous Cell Carcinoma (ESCC)	Tumour suppressor	Downregulated in tumour tissues; reduction correlates with advanced stage, poor differentiation, infiltration, and worse OS.	[61]
Gastric Cancer	Tumour promoter/Chemoresistance mediator	Upregulated in oxaliplatin-resistant tumours; silencing restores oxaliplatin sensitivity.	[62]
Pancreatic Ductal Adenocarcinoma (PDAC)	Context-dependent	In some PDACs, high CRABP2/low FABP5 expression correlates with retinoic acid sensitivity and anti-tumour effects. Other studies show CRABP2 is consistently overexpressed in PDAC, associated with poor prognosis, recurrence, and resistance to gemcitabine.	[63,64,65,66]
Hepatocellular Carcinoma (HCC)	Tumour promoter	Upregulated in tumours; promotes proliferation, migration, invasion, and angiogenesis via ERK–VEGF signalling.	[67]
Colorectal Cancer (CRC)	Dual role	Nuclear CRABP2 promotes proliferation and suppresses apoptosis via RB1 destabilisation. Cytoplasmic CRABP2 inhibits liver metastasis by activating mitophagy through AFG3L2–PINK1. CRABP2 also promotes peritoneal metastasis through EMT and invadopodia formation.	[68,69]

### 2.4. FABP1

FABP1 (liver FABP, L-FABP) is predominantly expressed in the liver and accounting for 10% of the total cytosolic protein with a dominant role in the binding, transport, and metabolism of long-chain fatty acids [70]. In addition to its primary ligands, FABP1 binds a variety of xenobiotics, including benzodiazepines, β-blockers, and non-steroidal anti-inflammatory drugs, helping to prevent their cytotoxic effects [71,72]. Several studies have shown that FABP1 expression is highly specific to tumours and is predominantly found in hepatocellular carcinomas (HCCs), colorectal carcinomas, and other gastrointestinal adenocarcinomas [73,74,75,76,77]. Different stages of colorectal tumour development show varying levels of L-FABP expression, with a notable loss at the adenoma stage with larger adenomas exhibiting significantly decreased FABP1 staining, and immunoreactivity is significantly associated with poorly differentiated cancers. FABP1 is also indicated as a marker of colorectal differentiation [75].

FABP1 is significantly downregulated in MSI (microsatellite instable) and medullary carcinomas, and its loss is associated with right-sided tumour location, high grade, and increased tumour-infiltrating lymphocytes. FABP1 expression showed a strong positive correlation with PPARγ, and both were reduced in the CMS1 (Consensus Molecular Subtypes) molecular subtype, which is characterised by immune infiltration and MSI. In vitro experiments indicated that interferon gamma (IFNγ), a key immune cytokine in MSI tumours, downregulates FABP1 expression, while activation of PPARγ (with rosiglitazone) can restore it, suggesting a regulatory loop (Figure 5) influenced by inflammatory signalling. The resulting disruption of the FABP1–PPARγ feedback loop may contribute to the distinct immune microenvironment of MSI tumours. Although FABP1 was not linked to prognosis in this study, its loss reflects broader immune-metabolic interactions characteristic of MSI CRC [78].

In HCC, FABP1 was shown to be a pro-tumourigenic factor that enhances angiogenesis and metastatic potential in HCC through upregulation of VEGF-A and its expression is significantly elevated in HCC tissues compared to adjacent normal tissues, with a strong positive correlation to VEGF-A levels. It was shown that FABP1 promotes VEGF-A expression via both direct transcriptional activation—mediated by the Akt/mTOR (mammalian Target of Rapamycin) in a Hypoxia Inducible Factor (HIF1a)-dependent way—and through interaction with VEGFR2 in cholesterol-rich membrane microdomains, activating downstream signalling pathways involving Src, Focal Adhesion Kinase (FAK), and cell division control protein 42 homolog (cdc42) that facilitate cell migration and neovascularisation. In vivo, FABP1 overexpression increased tumour growth, angiogenesis, and lung metastases in murine models, establishing its role as a promoter of tumour progression in HCC cells with high endogenous VEGF-A expression [79].

In contrast, Lin et al. characterize FABP1 as a tumour suppressor in HCC, where its downregulation—frequently observed in tumour tissues compared to adjacent non-tumourous liver—was associated with reduced lipid accumulation and poorer patient survival. FABP1 overexpression inhibited proliferation, migration, and invasion, and promoted apoptosis in HCC cells, especially in the presence of free fatty acids (FFAs), suggesting a protective role in maintaining lipid homeostasis (Table 5). Interleukin-6 (IL-6) induces the expression of miR-603, which in turn, inhibits the expression of FABP1 and enhances the expression of lipid metabolism-related oncogenic proteins [80].

**Table 5 curroncol-32-00531-t005:** Role of FABP1 in GI cancers.

Cancer Type	Role of FABP1	Details	Reference
Colorectal Cancer (CRC)	Tumour suppressor/Differentiation marker	Expression is reduced in adenomas and poorly differentiated cancers; significantly downregulated in MSI and medullary carcinomas. IFNγ downregulates FABP1 expression, while PPARγ activation restores it, suggesting a regulatory loop between inflammatory signalling and lipid metabolism.	[75,78]
Hepatocellular Carcinoma (HCC)	Context-dependent	FABP1 is significantly upregulated in some HCCs, promoting tumour growth, angiogenesis, and metastasis through VEGF-A transcription (via Akt/mTOR–HIF1α) and downstream signalling through VEGFR2 in cholesterol-rich domains. Other studies show downregulation of FABP1 in HCCs, associated with poor survival and reduced lipid accumulation. Overexpression inhibits proliferation, migration, and invasion, and promotes apoptosis.	[79,80]

### 2.5. FABP2

FABP2 (intestinal FABP, I-FABP) is abundantly expressed in the intestine, particularly at the tips of the villi, to serve as a key regulator of long-chain fatty acid uptake in intestinal enterocytes and has emerged as a potential biomarker and modulator in gastrointestinal cancers [2,81]. FABP2 is rapidly released into the bloodstream following mucosal tissue injury of the small intestine and it appears to be a useful biomarker for diagnosing acute intestinal ischemia [82].

In HCC, elevated FABP2 serum concentrations were observed in patients compared to healthy controls, suggesting increased intestinal permeability or enterocyte injury in the setting of chronic liver disease [83]. Despite its diagnostic relevance, FABP2 levels did not significantly correlate with overall survival, limiting its prognostic utility in HCC, since elevated levels likely reflect underlying cirrhosis-related gut–liver axis dysfunction rather than direct oncogenic processes [83].

In CRC, the Ala54Thr (rs1799883) polymorphism in FABP2 has been studied for its potential to modulate disease susceptibility via altered fatty acid uptake. In a Pakistani population, Ijaz et al. reported that the GG genotype conferred a significantly increased risk of CRC, supporting a potential role for this variant in tumourigenesis under specific genetic backgrounds [84]. However, findings from other populations have been less conclusive. The rs1799883 polymorphism alone was not significantly associated with CRC risk but may contribute through gene–gene and gene–environment interactions [85], or through a significant interaction between low dietary fat intake and the FABP2 Ala/Ala genotype, with effects more pronounced in proximal colon cancers (Table 6) [86].

**Table 6 curroncol-32-00531-t006:** Role of FABP2 in GI cancers.

Cancer Type	Role of FABP2	Details	Reference
Hepatocellular Carcinoma (HCC)	Biomarker	Elevated serum levels in patients compared to healthy controls; not significantly associated with overall survival.	[83]
Colorectal Cancer (CRC)	Genetic risk modifier	The Ala54Thr (rs1799883) polymorphism may modulate CRC susceptibility by affecting fatty acid uptake. In one population, the GG genotype was associated with increased risk; other studies found no direct association but suggest possible gene–environment interactions, particularly with dietary fat intake and proximal colon tumour risk.	[84,85,86]

### 2.6. FABP3

FABP3 (heart FABP, H-FABP) or mammary-derived growth inhibitor (MDGI) [87], may influence morphogenesis of the embryonic heart [88]. During terminal differentiation of cardiomyocytes, FABP3 is upregulated, inhibiting proliferation, promoting apoptosis, and affecting the differentiation of cardiac precursors into developed cardiomyocytes [89,90].

FABP3 has emerged as a consistent high-risk prognostic marker in gastric cancer (GC), identified across multiple studies through transcriptomic analyses, survival modelling, and functional validation. Its overexpression is linked to advanced disease stages, poor response to chemotherapy, and shorter overall survival [91,92]. FABP3 contributes to disease progression by supporting lipid metabolic reprogramming, particularly through the Regulation of Nuclear pre-mRNA Domain Containing 1B (RPRD1B)/c-Jun/c-Fos/SREBP1 axis, which enhances fatty acid uptake and metastasis [93]. Li et al. associated FABP3 with cuproptosis-related immune regulation, indicating its involvement in both metabolic and immunologic dysregulation in the tumour microenvironment [94]. Although most findings converge, there are minor differences in the magnitude of FABP3’s prognostic impact, with some studies identifying it as a modest but stable contributor in multigene signatures [94], while others highlight its strong individual prognostic significance [91,92].

In HCC, FABP3 plays a critical role in the development of sorafenib resistance and metastatic behaviour. It is significantly upregulated in sorafenib-resistant HCC cells and contributes to resistance by promoting fatty acid uptake, and the suppression of oxidative stress and apoptosis [95]. In vitro, FABP3 overexpression mitigates sorafenib-induced cell death, enhances mitochondrial stability, and reduces ROS production, whereas FABP3 knockdown restores drug sensitivity and increases apoptosis through Caspase-3 and Poly(ADP-ribose) polymerase (PARP) activation. FABP3 also drives HCC cell migration and invasion by activating the Phosphatidylinositol 3-kinase (PI3K)/AKT/Snail signalling pathway and promoting epithelial–mesenchymal transition. Treatment with oleanolic acid (OA) downregulates FABP3, sensitises resistant cells to sorafenib both in vitro and in vivo, and suppresses tumour growth and invasiveness [95].

In colon adenocarcinoma (COAD) FABP3 is downregulated and has been identified as an independent prognostic factor in a PPAR-related gene signature. Its reduced expression in tumour tissues and cancer cell lines suggests a potential tumour-suppressive role [96]. Although its individual predictive power is modest (Table 7), FABP3 is associated with fatty acid metabolism pathways and negatively correlates with immune cell infiltration, including dendritic and memory CD4+ T cells [96].

**Table 7 curroncol-32-00531-t007:** Role of FABP3 in GI cancers.

Cancer Type	Role of FABP3	Details	Reference
Gastric Cancer (GC)	Tumour promoter/Prognostic marker	Overexpressed in advanced stages; associated with poor chemotherapy response and reduced overall survival. Linked to immune dysregulation via cuproptosis-related pathways.	[91,92,93,94]
Hepatocellular Carcinoma (HCC)	Tumour promoter/Drug resistance mediator	Upregulated in sorafenib-resistant cells; promotes resistance by enhancing fatty acid uptake, stabilizing mitochondria, and reducing ROS.	[95]
Colon Adenocarcinoma (COAD)	Tumour suppressor	Downregulated in tumours and cell lines; identified as an independent prognostic factor in a PPAR-related gene signature.	[96]

### 2.7. FABP4

FABP4 (adipocyte FABP, A-FABP), is mainly expressed in adipocytes and macrophages, playing a crucial role in systemic metabolic regulation through multiple signalling pathways [97]. It serves as a significant prognostic biomarker for metabolic disorders, such as obesity, metabolic syndrome, type 2 diabetes, and atherosclerosis, detected at elevated levels in serum [97,98]. In the context of cancer, FABP4 is crucial in driving tumour proliferation, metastasis, and drug resistance by activating oncogenic signalling pathways and modulating cellular metabolism to meet energy demands, highlighting its potential as a therapeutic target in cancer treatment [99].

In CRC, FABP4 is a key regulator, contributing to tumour progression, metastasis, metabolic reprogramming, and drug resistance. Across multiple studies, FABP4 is consistently reported to be significantly upregulated in CRC tissues and cell lines, and its expression correlates with aggressive clinicopathological features [100,101,102]. Several studies have shown that FABP4 promotes the epithelial–mesenchymal transition (EMT) by downregulating E-cadherin and upregulating mesenchymal markers, such as Snail, MMPs, N-cadherin, and Vimentin, thereby enhancing invasion and metastatic potential [100,103,104]. FABP4 also fosters lipid metabolic reprogramming in CRC, increasing lipid droplet formation through upregulation of Fatty Acid Synthase (FASN), stearoyl-CoA desaturase (SCD), and acyl-CoA synthetase long-chain family member 1 (ACSL1), and contributing to elevated energy metabolism and glycolysis, marked by higher extracellular acidification rate (ECAR), lactate production, and glucose transporter 1 (Glut1)/lactate dehydrogenase A (LDHA) expression [101,103,104]. Notably, Zuo et al. showed that tripartite motif-containing protein 3 (TRIM3) reversed these oncogenic functions by promoting FABP4 ubiquitination and degradation, thereby reversing its pro-metastatic and lipogenic effects [104].

Recent evidence also revealed FABP4 as a critical mediator of treatment resistance in CRC, particularly against anti-EGFR therapies. Cheng et al. demonstrated that the FABP4/uncoupling protein 2 (UCP2) axis drives resistance to cetuximab in drug-tolerant persister cells and KRAS-mutated CRC models, primarily by supporting metabolic dormancy and adipocyte-induced lipid remodelling [105]. This FABP4/UCP2 axis was enriched in adipocyte-rich tumour microenvironments and correlated with increased stemness and mesenchymal traits in persisted cells, which could be reversed by the FABP4 inhibitor BMS309403 [105]. Although most studies highlight the oncogenic role of cytoplasmic FABP4, Kim et al. noted that nuclear expression of FABP4 did not correlate with poor survival outcomes, suggesting subcellular localisation might influence FABP4’s functional impact [102]. Taken together, these studies identify FABP4 as a key molecular driver of CRC aggressiveness and treatment resistance, with consistent evidence supporting its involvement in EMT, lipid metabolism, and therapeutic tolerance, and propose it as a promising therapeutic target for CRC, particularly in obesity-associated and drug-refractory settings.

FABP4 also promotes pancreatic cancer progression by enhancing tumour cell proliferation and supporting redox homeostasis [106]. Exogenous FABP4 significantly increased the proliferation of PDAC cell lines (Panc-1 and Pan02) by facilitating the G1-to-S/G2 phase transition. While FABP4 did not significantly affect apoptosis, it upregulated endogenous FABP4 expression and activated the nuclear factor erythroid 2-related factor 2 (Nrf2) pathway, leading to reduced intracellular ROS levels. In vivo, tumour growth was significantly reduced in FABP4 knockout mice, indicating that host-derived FABP4 supports PDAC tumour development [106].

In HCC, FABP4 appears to play a dual role, with its function influenced by the metabolic context. FABP4 is downregulated in HCC tissues and functions as a tumour suppressor [107]. Overexpression of FABP4 in HCC cell lines inhibited proliferation and migration in vitro and suppressed tumour growth in vivo, while FABP4 knockdown showed the opposite effect. FABP4 may exert its tumour-suppressive effects by downregulating Snail and phosphorylated STAT3 (p-STAT3), inhibiting the epithelial-mesenchymal transition (EMT) and oncogenic STAT3 signalling [107]. Low FABP4 expression was associated with more aggressive tumour features and poor prognosis, establishing it as an independent prognostic factor [107]. Conversely, FABP4 is markedly upregulated in obesity-associated HCC, particularly in response to high-fat diets and fatty acid exposure. In this context, exogenous FABP4 promoted HCC cell proliferation and migration, indicating a pro-tumourigenic role. These contrasting findings suggest that FABP4 may exert tumour-suppressive effects in general HCC, but facilitate tumour progression in obesity-related HCC, highlighting its context-dependent function [108].

FABP4 is overexpressed in oral squamous cell carcinoma (OSCC) of the tongue, with positive staining observed in tumour regions but not in adjacent non-tumour tissues. In vitro, FABP4-specific siRNA in SAS (cell line) tongue SCC cells demonstrated that FABP4 knockdown significantly reduced cell growth in a dose-dependent manner by decreasing the expression and phosphorylation of MAPK (Table 8). These findings suggest that FABP4 promotes OSCC progression, at least in part, through MAPK pathway activation, and may serve as a potential therapeutic target in tongue SCC [109].

**Table 8 curroncol-32-00531-t008:** Role of FABP4 in GI cancers.

Cancer Type	Role of FABP4	Details	Reference
Colorectal Cancer (CRC)	Tumour promoter	Upregulated in tumour tissues; associated with EMT, metastasis, lipid metabolic reprogramming, and drug resistance. Subcellular localisation may influence function.	[100,101,102,103,104,105]
Pancreatic Ductal Adenocarcinoma (PDAC)	Tumour promoter	Enhances proliferation by promoting cell cycle progression and activating the Nrf2 pathway, which reduces ROS levels.	[106]
Hepatocellular Carcinoma (HCC)	Context-dependent	Downregulated in general HCC; overexpression inhibits proliferation, migration, EMT, and STAT3 signalling, suggesting tumour-suppressive function. Upregulated in obesity-associated HCC and promotes proliferation and migration in response to high-fat diets and fatty acids.	[107,108]
Oral Squamous Cell Carcinoma (OSCC, tongue)	Tumour promoter	Overexpressed in tumour tissues; knockdown reduces proliferation by suppressing MAPK signalling.	[109]

### 2.8. FABP5

FABP5 (epidermal FABP, keratinocyte FABP, cutaneous FABP, C-FABP), or psoriasis-associated FABP (PA-FABP), plays crucial roles in fatty acid binding, trafficking, lipid metabolism, and cell growth [110]. Specifically, C-FABP is vital for FA transport and metabolism in the epidermis, and its alteration can affect the proliferation of keratinocytes [111]. Psoriatic skin and oral mucosa exhibit higher metabolism and transport for FAs than normal epidermis [112].

In gastric cancer, FABP5 has been identified as an oncogenic driver. Several studies report significantly elevated FABP5 expression in GC tissues compared to normal mucosa, correlating with adverse clinical features such as poor differentiation, lymph node metastasis, vascular invasion, and shortened overall survival [113,114,115,116]. Functionally, FABP5 enhances GC cell proliferation, migration, invasion, and survival, while its silencing or pharmacologic inhibition (e.g., with SBFI-26 (1β,2R,3α,4R)-2,4-diphenyl-1,3-cyclobutanedicarboxylic acid, 1-(1-naphthalenyl) ester)) induces apoptosis and suppresses proliferation both in vitro and in vivo [114,116].

FABP5 activates YAP1 via the Hippo signalling pathway, promoting transcription of proliferation-related genes such as the connective tissue growth factor (*CTGF)* and the cysteine-rich angiogenic inducer 61(*CYR61*) [116]. Additionally, FABP5 contributes to immune evasion by positively regulating the programmed cell death ligand 1 (PD-L1), the tumour necrosis factor (TNF), and IL-17 expression, and reshaping the tumour immune microenvironment and facilitating immune escape [115]. In lipid-rich microenvironments, like the omentum, FABP5 mediates palmitic acid-induced metastasis by translocating to the nucleus and activating the SP1/UCA1 axis, a pathway linked to enhanced GC cell migration and invasion [113].

Multiple studies have shown that FABP5 is significantly overexpressed in HCC tissues compared to adjacent non-tumourous liver and correlates with aggressive clinical features, such as poor differentiation, microvascular invasion, larger tumour size, and reduced overall and disease-free survival [117,118,119,120]. FABP5 enhances tumour cell proliferation and migration via pathways including PI3K/AKT/mTOR and cAMP response element-binding protein (CREB)-mediated miR-889-5p signalling, which targets and suppresses the tumour suppressor Kruppel-like factor 9 (KLF9) [121]. Additionally, FABP5 contributes to angiogenesis by activating the IL6/STAT3/VEGFA axis [117] and facilitates EMT by upregulating Snail and promoting β-catenin nuclear translocation [118].

FABP5 also shapes the tumour microenvironment. It reprograms tumour-associated monocytes to adopt a lipid-accumulating, immunosuppressive phenotype, enhancing PD-L1 expression on Treg cells through IL-10–JNK–STAT3, thus fostering immune evasion [122]. It has been also shown that FABP5 supports tumour survival under hypoxic or nutrient-deprived conditions via a FABP5–HIF-1α axis that upregulates both hypoxia-responsive and lipid metabolism genes [123]. Genomic evidence further substantiates its role, with frequent amplification of the FABP5 locus and associated co-expression of proliferation-associated genes such as cyclin B1 (*CCNB1*) and Polo-like kinase 1 (*PLK1*) in HCC [120]. Across studies, FABP5 inhibition consistently attenuated malignant phenotypes, and FABP5-high HCC cells showed sensitivity to the FABP5 inhibitor SBFI-26, underscoring its potential as a robust therapeutic target in HCC [117,120,123].

FABP5 plays an oncogenic role in pancreatic neuroendocrine neoplasms (pNENs), where it is consistently overexpressed in tumour tissues and cell lines and promotes proliferation, migration, invasion, and tumour growth both in vitro and in vivo [124,125]. FABP5 exerts these effects through stabilisation of fatty acid synthase (FASN), enhancing lipid accumulation and cholesterol synthesis, which are essential for Wnt pathway activation [124]. Furthermore, Chen et al. identified FABP5 as a downstream effector of the m6A demethylase ALKBH5, which increases FABP5 mRNA stability via an insulin-like growth factor 2 mRNA binding protein 2 (IGF2BP2)-dependent mechanism. FABP5 also activates the PI3K/Akt/mTOR pathway, further promoting tumourigenesis [125]. In addition to its direct effects in pNEN cells, FABP5 can influence tumour progression indirectly. In tumour-associated macrophages, FABP5 interacts with RTN3 to regulate lipid metabolism. Under hypoxia, exosomal miR-4488 from pNEN cells suppresses RTN3, which in turn downregulates FABP5, promoting M2 macrophage polarisation and activating the PI3K/Akt/mTOR pathway. This immunosuppressive shift enhances MMP2 secretion and supports tumour metastasis, while restoring FABP5 reverses these effects [126].

FABP5 emerges as a promoter of colorectal cancer (CRC) progression. Across multiple studies, FABP5 is consistently reported to be overexpressed in CRC tissues and cell lines, with elevated expression correlating with increased tumour cell proliferation, migration, and invasiveness [127,128,129]. FABP5 facilitates tumour growth by modulating key signalling pathways, such as PI3K/AKT, MAPKs, and NF-κB, particularly within inflammatory and lipid-rich tumour microenvironments mimicked by adipocyte-conditioned medium (aCM) [127]. FABP5 also enhances the expression of hypoxia-inducible factor 1-alpha (HIF-1α) and its downstream targets, promoting CRC cell adaptation and growth under hypoxic conditions [129]. FABP5 promotes lipid metabolism by interacting with lipid-processing enzymes, such as the monoacylglycerol lipase (MAGL) and the hormone-sensitive lipase (HSL), and by upregulating genes, like Acetyl-CoA carboxylase alpha (ACCα) and isocitrate dehydrogenase (NADP(+)) 1 (IDH1), thereby sustaining the metabolic demands of rapidly proliferating cells [128]. Although PPARβ/δ has been implicated in FABP5-mediated oncogenic signalling in other cancers, its role appears to be non-essential in CRC, with FABP5-driven progression occurring through alternative pathways [128]. Various mechanisms for FABP5 are summarised in Figure 6 and Table 9.

**Table 9 curroncol-32-00531-t009:** Role of FABP5 in GI cancers.

Cancer Type	Role of FABP5	Details	Reference
Gastric Cancer (GC)	Tumour promoter	Overexpressed in tumour tissues; associated with poor differentiation, lymph node metastasis, vascular invasion, and reduced survival. Promotes proliferation, invasion, and survival. Enhances immune evasion by upregulating PD-L1, TNF, and IL-17.	[113,114,115,116]
Hepatocellular Carcinoma (HCC)	Tumour promoter	Strongly overexpressed; correlates with aggressive features and poor survival. Shapes immunosuppressive microenvironment by inducing lipid-accumulating monocytes and PD-L1^+^ Tregs.	[117,118,119,120,121,122,123]
Pancreatic Neuroendocrine Neoplasms (pNENs)	Tumour promoter	Consistently overexpressed; enhances tumour growth, migration, and invasion by stabilizing FASN and supporting cholesterol synthesis.	[124,125]
Colorectal Cancer (CRC)	Tumour promoter	Overexpressed in CRC tissues and cell lines; promotes proliferation, migration, and invasion via PI3K/AKT, MAPKs, NF-κB, and HIF-1α signalling.	[127,128,129]

### 2.9. FABP6

FABP6 (ileal-FABP), is associated with the bile acid transport in ileal epithelial cells and is crucial for efficient digestion and absorption of dietary fats. Bile acids have been reported to induce colon carcinogenesis inducing colonic epithelial cells inflammation, resulting in genetic modulation through DNA damage [130,131,132]. Additionally, abnormal levels of bile acids are implicated in hepatic inflammation and tumourigenesis [133]. Due to its role as transporter, FABP6 plays an essential role in CRC [134].

In CRC, FABP6 overexpression might be associated with early-phase carcinogenesis, as higher levels of FABP6 correlate with smaller tumour size, more frequent location in the left colon, and reduced depth of tumour invasion. Moreover, metastatic cells and lesions derived from lymph nodes exhibited lower levels of FABP6 mRNA expression [135]. A later study supported these findings, revealing that FABP6 expression was upregulated by about 10-fold in the sessile serrated adenomas, which are precursors to CRC, in comparison to normal colonic tissue, and by more than 20-fold compared to benign polyps [136].

A novel transcript of FABP6 has been reported to potentially protect colon cancer cells from apoptosis via nuclear factor (NF)-kB pathway activation [132]. Lastly, FABP6 may serve as a favourable prognostic biomarker for PDAC, as indicated by the multivariate Cox proportional hazards regression analysis (Table 10) [137].

**Table 10 curroncol-32-00531-t010:** Role of FABP6 in GI cancers.

Cancer Type	Role of FABP6	Details	Reference
Colorectal Cancer (CRC)	Early-stage tumour marker/Potential tumour promoter	Overexpressed in early lesions such as sessile serrated adenomas; expression correlates with smaller tumour size, left-sided location, and reduced invasion depth.	[132,134,135,136]
Pancreatic Ductal Adenocarcinoma (PDAC)	Prognostic biomarker	Identified as a favourable prognostic marker.	[137]

### 2.10. FABP7

FABP7 (brain lipid binding protein, BLBP), was initially identified in a foetal brain cDNA library. It is predominantly found in the adult human brain and skeletal muscle, with minimal expression in other normal adult tissues. Its expression is particularly high during the immature stages of brain development and decreases as the brain matures [138]. Notably, FABP7 is overexpressed in various cancers, including melanomas and glioblastomas, influencing cell proliferation, and its levels are associated with patient outcomes [139,140].

FABP7 is significantly upregulated in CRC tissues and cell lines compared to adjacent non-tumourous tissues [141]. In vitro, FABP7 enhances CRC cell proliferation and inhibits apoptosis via activation of the MEK/ERK signalling pathway, while its knockdown suppressed tumour growth in vivo and in vitro, implicating it as a potential therapeutic target in CRC pathophysiology (Table 11) [141].

**Table 11 curroncol-32-00531-t011:** Role of FABP7 in GI cancers.

Cancer Type	Role of FABP7	Details	Reference
Colorectal Cancer (CRC)	Tumour promoter	Significantly upregulated in CRC tissues and cell lines; promotes proliferation and inhibits apoptosis via activation of the MEK/ERK pathway.	[141]

## 3. Concluding Remarks

Intracellular lipid-binding proteins (iLBPs), including CRBPs, CRABPs, and FABPs, are central regulators of retinoid and lipid signalling, integrating metabolic and transcriptional control to influence cancer cell phenotypes. In gastrointestinal (GI) cancers, dysregulation of these proteins contributes to key oncogenic processes: ranging from altered differentiation and proliferation to immune evasion and therapy resistance.

However, most of the experimental evidence underpinning the involvement of iLBPs in drug resistance and immune evasion derives from in vitro cell culture models, in vivo murine studies, and patient-derived xenografts, with limited direct validation in clinical samples [62,63,64,65,95,105]. While associations between iLBP expression and prognosis or therapeutic response have been consistently reported across tumour cohorts [73,74,75,76,77,83,91,92,93,94,100,101,102,103,104,105,117,118,119,120,121,122,123], these remain largely retrospective or bioinformatics-based analyses, and prospective clinical validation is required before their use as biomarkers can be translated into routine clinical practice.

An emerging challenge, considering iLBPs as potential therapeutic targets, is to achieve isoform specificity given the high degree of structural homology within the FABP family, and to mitigate off-target effects given their broad physiological functions. However, early preclinical studies of small-molecule inhibitors (e.g., FABP4 inhibitor BMS309403, FABP5 inhibitor SBFI-26, CRABP2 degrader SNIPER-11) have demonstrated proof-of-principle efficacy [65,105,116]. In addition, natural compounds, such as oleanolic acid and pterostilbene, have also been shown to modulate FABP3- and FABP5-related oncogenic pathways, restoring drug sensitivity and suppressing tumour cell migration, respectively [95,127]. These findings highlight the potential of dietary or plant-derived agents to serve as complementary approaches to pharmacological strategies.

Finally, considering the role of ion channels in GI cancers, it is important to consider how iLBPs may interact with them. Ion channels are well known to be modulated by their lipid environment through both non-specific effects (membrane curvature, tension, hydrophobic mismatch) and specific lipid–protein interactions with cholesterol, polyunsaturated fatty acids, and phosphoinositides [142]. iLBPs are well positioned to influence channel activity by controlling the intracellular trafficking and availability of these lipid ligands. Consistent with this, in neuronal models FABP5 deletion increases the intracellular levels of endocannabinoids/NAE, thereby enhancing signalling through CB1 and PPARα receptors and ultimately suppressing NGF-driven TRPV1 sensitisation [143]. Similarly, a FABP3 ligand (MF1) can potentiate GABAA currents by acting at the benzodiazepine recognition site leading to anti-seizure effects in vivo. These effects can be reversed by flumazenil confirming that MF1 acts as an allosteric modulator of the receptor [144]. In addition, FABP7 has been shown to influence glutamatergic ion channels. When FABP7 is deleted, docosahexaenoic acid (DHA) no longer enhances NMDA receptor activity in hippocampal neurons [145]. FABP7 deletion is also linked to greater NMDA receptor availability in striatal and cortical regions and mice lacking FABP7 show stronger locomotor responses when treated with the NMDA receptor blocker MK-801 [146,147]. These studies show mechanisms by which iLBPs can directly or indirectly affect ion channel function. Although direct experimental evidence in GI cancers is currently lacking, these mechanistic studies suggest that iLBP-mediated lipid trafficking and ligand delivery could critically influence ion channel signalling, thereby opening a promising new direction for future research in GI cancer research.

Overall, key mechanistic studies as well as clinical validation are required to better understand and leverage the role of iLBPs in cancer development and progression for therapeutic benefit.

## Figures and Tables

**Figure 1 curroncol-32-00531-f001:**
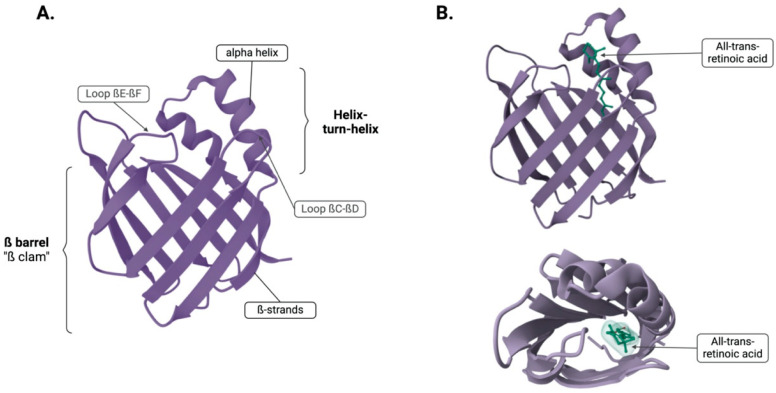
(**A**) Tertiary structure showing the ß-barrel and the two alpha helices. (**B**) Two different views of CRABP2 with the ligand atRA bound. Created in BioRender. Kakouratos, C. (2025) https://BioRender.com/hkfanlq and structures from Protein Data Bank (PDB).

**Figure 2 curroncol-32-00531-f002:**
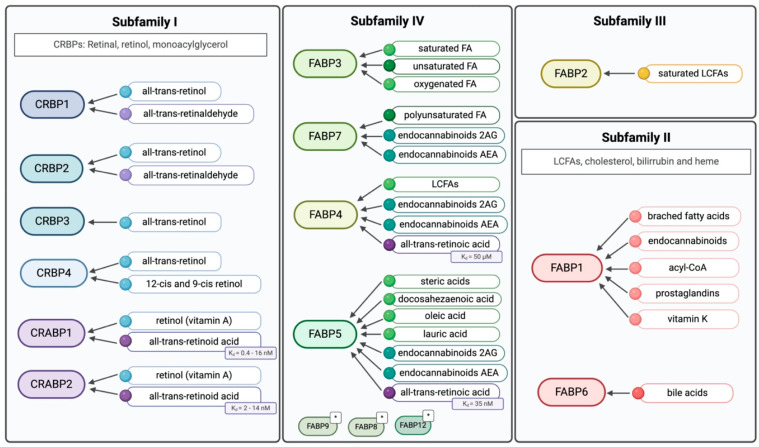
Ligands for the four different iLBPs subfamilies. Subfamilies closer to each other represents their similarity, as well as colours within the subfamilies. * Their ligand-binding properties have not been extensively characterised. Created in BioRender. Kakouratos, C. (2025) https://BioRender.com/rhjvzv0.

**Figure 3 curroncol-32-00531-f003:**
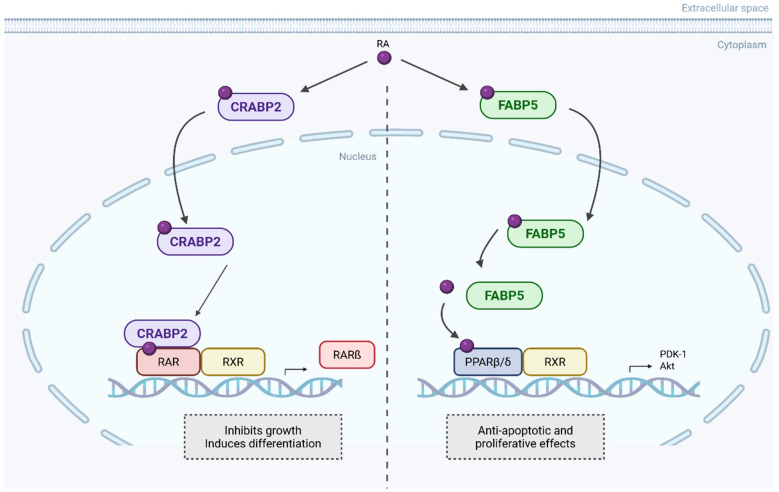
CRABP2 and FABP5 opposite pathways upon RA binding. RA is transported to the nucleus by CRABP2 and binds to retinoic acid receptor α (RARα), forming heterodimers. This classical RA pathway leads to gene expression that inhibits cell growth. Alternatively, RA can be transported by FABP5 to non-classical receptors, such as PPARβ/δ, activating non-genomic pathways that promote cell proliferation and inhibit apoptosis. Created in BioRender. Kakouratos, C. (2025) https://BioRender.com/qxxrpl6. Adapted from Gupta et al. [63].

**Figure 4 curroncol-32-00531-f004:**
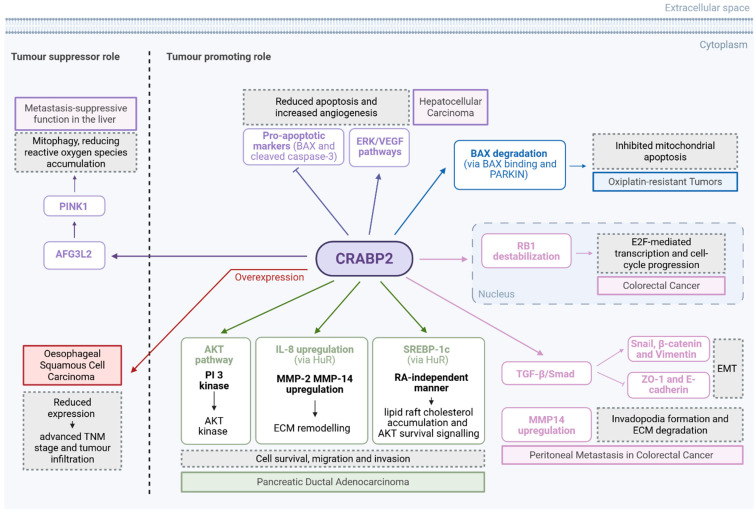
The role of CRABP2 as tumour suppressor and tumour promoter via different pathways in GI cancers. Created in BioRender. Kakouratos, C. (2025) https://BioRender.com/471o42b.

**Figure 5 curroncol-32-00531-f005:**
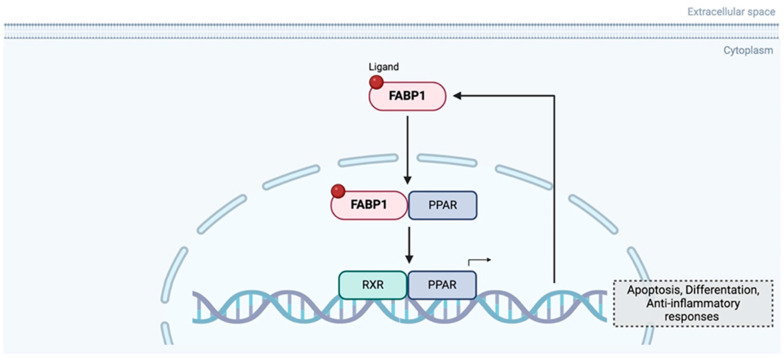
Feedback regulator loop between FABP1 and PPAR. Created in BioRender. Kakouratos, C. (2025) https://BioRender.com/rwq0szb.

**Figure 6 curroncol-32-00531-f006:**
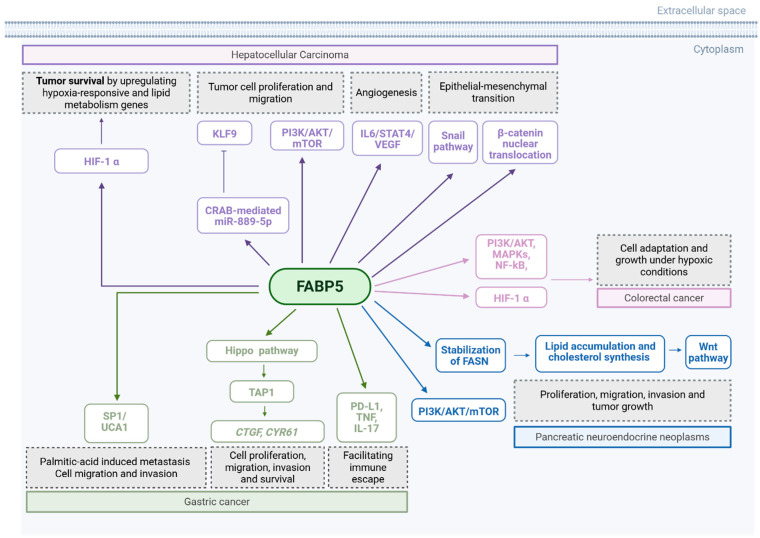
Summary mechanisms by which FABP5 promotes tumour growth and metastasis. EGFR: endothelial growth factor receptor. VEGF: vascular endothelial growth factor. Created in BioRender. Kakouratos, C. (2025) https://BioRender.com/4r75vj9.

**Table 1 curroncol-32-00531-t001:** Summary of iLBPs properties. RPKM: Read per Kilobase of transcript per Million mapped reads. ^(a)^ Molecular weight from (4). ^(b)^ Gene location obtained from the National Center of Biotechnology and Information gene database.

Protein Name *(Gene Name)*	Tissue-Specific Name *(Gene Name)*	Sub-Family	Molecular Weight (kDa) ^(a)^	Chromo-Somal Location ^(b)^	Expression and Distribution (Biased, More Expressed in (RPKM)) ^(b)^
CRBP1	*RBP1*	I	15.9	3q23	Ovary (127.5), adrenal (41.0), endometrium, gall bladder, testis, liver, placenta, and pancreas.
CRBP2	*RBP2*	I	15.7	3q23	Duodenum (1065.7) and small intestine (769.4).
CRBP3	*RBP5*	I	15.9	12p13.31	Kidney (131.3), liver (58.1), spleen, and lymph node
CRBP4	*RBP7*	I	15.5	1p36.22	Fat (138.2), spleen (25.4), gall bladder, thyroid, heart, kidney, placenta, endometrium
CRABP1	*CRABPI*	I	15.6	15q25.1	Thyroid (65.7), spleen (23.2), placenta, skin, kidney, brain
CRABP2	*RBP6*	I	15.7	1q23.1	Oesophagus (46.7), skin (7.1), endometrium, urinary bladder, placenta
*FABP1*	Liver *(L-FABP)*	II	14.2	2p11.2	Liver (2613.3), colon (1932.3), duodenum, small intestine, kidney
*FABP2*	Intestinal *(I-FABP)*	III	14.4	4q26	Small intestine (287), duodenum (142.3), and colon
*FABP3*	Heart *(H-FABP)*	IV	15.2	1p35.2	Heart (311.6), brain (22.8), kidney, and prostate
*FABP4*	Adipocyte *(A-FABP)*	IV	14.9	8q21.13	Fat (3748.8) and placenta (92.5)
*FABP5*	Epidermal *(K-FABP)*	IV	14.7	8q21.13	Oesophagus (725.7), fat (171.7), placenta, skin, lung, colon, appendix, lymph node, stomach
*FABP6*	Ileal *(Il-FABP)*	II	15.2	5q33.3	Small intestine (820.3)
*FABP7*	Brain *(B-FABP)*	IV	14.9	6q22.31	Brain (8.9) and skin (0.4)
*FABP8*	*PMP2*	IV	14.9	8q21.13	Brain (121.5)
*FABP9*	Testis *(T-FABP)*	IV	14.1	8q21.13	Fat (0.3)—low expression in dataset
*FABP12*	*FABP12*	IV	15.6	8q21.13	Testis (0.6)—low expression in dataset

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
