# Peer review of "The Role of Intracellular Lipid-Binding Proteins in Digestive System Neoplasms"

_curroncol, 2025, doi:10.3390/curroncol32100531_

Round 1

Reviewer 1 Report

Comments and Suggestions for Authors

Intracellular lipid-binding proteins (iLBPs) are crucial mediators of intracellular transport for fatty acids and retinoids, functioning as lipid chaperones. In addition to lipid transport, iLBPs play significant roles in regulating signaling pathways, gene expression, oxidative balance, and inflammation. They are increasingly recognized for their involvement in gastrointestinal (GI) diseases, particularly in cancer. iLBPs are categorized into four distinct subfamilies, each exhibiting unique tissue distributions and ligand preferences. iLBPs extend beyond lipid trafficking to intersect with oncogenic pathways, influencing cell fate and treatment responses, thereby underscoring their potential as biomarkers and therapeutic targets in GI oncology. However, some issues should be resolved before accepted.

1.Papers about FABP5, such as, PMID: 38904015, should be cited.

2.Title: The Role of Intracellular Lipid Binding Proteins in Gastrointestinal Cancers should be revised Digestive System Neoplasms‌

Author Response

Intracellular lipid-binding proteins (iLBPs) are crucial mediators of intracellular transport for fatty acids and retinoids, functioning as lipid chaperones. In addition to lipid transport, iLBPs play significant roles in regulating signaling pathways, gene expression, oxidative balance, and inflammation. They are increasingly recognized for their involvement in gastrointestinal (GI) diseases, particularly in cancer. iLBPs are categorized into four distinct subfamilies, each exhibiting unique tissue distributions and ligand preferences. iLBPs extend beyond lipid trafficking to intersect with oncogenic pathways, influencing cell fate and treatment responses, thereby underscoring their potential as biomarkers and therapeutic targets in GI oncology. However, some issues should be resolved before accepted.

1.Papers about FABP5, such as, PMID: 38904015, should be cited.

The paper with PMID: 38904015 has now been cited in the FABP5 section (page 17, line 593, Ref 141).

2.Title: The Role of Intracellular Lipid Binding Proteins in Gastrointestinal Cancers should be revised Digestive System Neoplasms‌

The manuscript title has been revised to “The Role of Intracellular Lipid Binding Proteins in Digestive System Neoplasms.”

Reviewer 2 Report

Comments and Suggestions for Authors

This review article provides a comprehensive and well-structured overview of the role of intracellular lipid-binding proteins (iLBPs) in gastrointestinal cancers. The authors clearly summarize the structural and functional diversity of CRBPs, CRABPs, and FABPs, and their context-dependent oncogenic or tumor-suppressive effects. The manuscript is well written, logically organized, and rich in mechanistic detail, which will be of interest to both basic researchers and clinicians. Figures and tables are informative and enhance readability. I suggest several minor issues which need to be addressed before acceptance.

Page 1, lines 38: Please use “Intracellular lipid-binding protein” with a hyphen consistently.

Page 4, Table 1: Please remove duplicated word “thyroid” in Expression and distribution of CRBP4.

Page 5, line 146: Please correct grammar: change “forms complex lipids” to “form complex lipids.”

Page 7, line 202: Please remove extra space before the period in “ESCC .”

Page 9, line 322: Please correct grammar: “when is localized” to “when it is localised.”

Page 12, line 399, Table 1: Please correct table numbering; change “Table 1. Role of FABP1 in GI cancers.” to “Table 5.”

Page 15, line 531, Table 2: Please correct table numbering; change “Table 2. Role of FABP4 in GI cancers.” to “Table 8.”

Page 19, line 640: Please correct typo: “it’s knockdown” to “its knockdown.”

Author Response

This review article provides a comprehensive and well-structured overview of the role of intracellular lipid-binding proteins (iLBPs) in gastrointestinal cancers. The authors clearly summarize the structural and functional diversity of CRBPs, CRABPs, and FABPs, and their context-dependent oncogenic or tumor-suppressive effects. The manuscript is well written, logically organized, and rich in mechanistic detail, which will be of interest to both basic researchers and clinicians. Figures and tables are informative and enhance readability. I suggest several minor issues which need to be addressed before acceptance.

Page 1, lines 38: Please use “Intracellular lipid-binding protein” with a hyphen consistently.

Page 1, line 38: A hyphen has now been added to “Intracellular lipid-binding protein” for consistency.

Page 4, Table 1: Please remove duplicated word “thyroid” in Expression and distribution of CRBP4.

Page 4, Table 1: The duplicated word “thyroid” has been removed.

Page 5, line 146: Please correct grammar: change “forms complex lipids” to “form complex lipids.”

Page 5, line 146: “forms complex lipids” has been corrected to “form complex lipids.”

Page 7, line 202: Please remove extra space before the period in “ESCC .”

Page 7, line 202: The extra space before the period in “ESCC .” has been removed.

Page 9, line 322: Please correct grammar: “when is localized” to “when it is localised.”

Page 9, line 322: “when is localized” has been corrected to “when it is localised.”

Page 12, line 399, Table 1: Please correct table numbering; change “Table 1. Role of FABP1 in GI cancers.” to “Table 5.”

Page 12, line 399: The table number has been changed from “Table 1. Role of FABP1 in GI cancers.” to “Table 5.”

Page 15, line 531, Table 2: Please correct table numbering; change “Table 2. Role of FABP4 in GI cancers.” to “Table 8.”

Page 15, line 531: The table number has been changed from “Table 2. Role of FABP4 in GI cancers.” to “Table 8.”

Page 19, line 640: Please correct typo: “it’s knockdown” to “its knockdown.”

Page 19, line 647: “it’s knockdown” has been corrected to “its knockdown.”

We sincerely thank the reviewer for their thoughtful and precise comments, which have helped us refine the manuscript and improve its overall clarity.

Reviewer 3 Report

Comments and Suggestions for Authors

The manuscript provides an overview of intracellular lipid-binding proteins (iLBPs) and their roles beyond lipid transport, highlighting their involvement in signaling, oxidative balance, inflammation, and particularly gastrointestinal oncology. The topic is timely and relevant, as iLBPs are emerging as potential biomarkers and therapeutic targets in cancer. The abstract effectively emphasizes the dual and context-dependent functions of these proteins, as well as their implications for drug resistance and immune evasion.

While the manuscript addresses an important subject, there are several areas where further clarification and discussion would strengthen the work.

On what type of experimental evidence (e.g., in vitro, in vivo, or patient-derived data) is the claim regarding iLBPs’ involvement in drug resistance and immune evasion based?

What is the current level of clinical validation for iLBPs as potential biomarkers in gastrointestinal oncology?

When considering iLBPs as therapeutic targets, what are the main challenges in terms of specificity and potential off-target effects?

Have there been studies investigating the modulation of iLBPs by natural products or plant-derived compounds, and if so, could the authors discuss their potential therapeutic implications?

Considering that numerous ion channels are implicated in gastrointestinal cancers, it would be valuable if the authors could elaborate on how iLBPs may interact with or influence ion channel–mediated oncogenic pathways.

Author Response

The manuscript provides an overview of intracellular lipid-binding proteins (iLBPs) and their roles beyond lipid transport, highlighting their involvement in signaling, oxidative balance, inflammation, and particularly gastrointestinal oncology. The topic is timely and relevant, as iLBPs are emerging as potential biomarkers and therapeutic targets in cancer. The abstract effectively emphasizes the dual and context-dependent functions of these proteins, as well as their implications for drug resistance and immune evasion.

While the manuscript addresses an important subject, there are several areas where further clarification and discussion would strengthen the work.

We sincerely thank the reviewer for their thoughtful and constructive comments, which have helped us strengthen the depth and scope of our manuscript. In response, we have expanded the Concluding Remarks section to address all the reviewer’s points in greater detail.

On what type of experimental evidence (e.g., in vitro, in vivo, or patient-derived data) is the claim regarding iLBPs’ involvement in drug resistance and immune evasion based?

We now clarify the types of experimental evidence (in vitro, in vivo, and patient-derived data) supporting iLBPs’ involvement in drug resistance and immune evasion.

However, most of the experimental evidence underpinning the involvement of iLBPs in drug resistance and immune evasion derives from in vitro cell culture models, in vivo murine studies, and patient-derived xenografts, with limited direct validation in clinical samples [62–65,95,105].

What is the current level of clinical validation for iLBPs as potential biomarkers in gastrointestinal oncology?

We provide a discussion on the current level of clinical validation for iLBPs as potential biomarkers in gastrointestinal oncology.

While associations between iLBP expression and prognosis or therapeutic response have been consistently reported across tumour cohorts [73–77,83,91–94,100–105,117–123], these remain largely retrospective or bioinformatics-based analyses, and prospective clinical validation is required before their use as biomarkers can be translated into routine clinical practice.

When considering iLBPs as therapeutic targets, what are the main challenges in terms of specificity and potential off-target effects?

We highlight the main challenges of targeting iLBPs, particularly regarding specificity and off-target effects.

An emerging challenge, considering iLBPs as potential therapeutic targets, is to achieve isoform specificity given the high degree of structural homology within the FABP family, and to mitigate off-target effects given their broad physiological functions. However, early preclinical studies of small-molecule inhibitors (e.g., FABP4 inhibitor BMS309403, FABP5 inhibitor SBFI-26, CRABP2 degrader SNIPER-11) have demonstrated proof-of-principle efficacy [65,105, 116].

Have there been studies investigating the modulation of iLBPs by natural products or plant-derived compounds, and if so, could the authors discuss their potential therapeutic implications?

We include a small discussion of studies investigating the modulation of iLBPs by natural and plant-derived compounds, outlining their potential therapeutic implications.

In addition, natural compounds such as oleanolic acid and pterostilbene have also been shown to modulate FABP3- and FABP5-related oncogenic pathways, restoring drug sensitivity and suppressing tumour cell migration, respectively [95,126]. These findings highlight the potential of dietary or plant-derived agents to serve as complementary approaches to pharmacological strategies.

Considering that numerous ion channels are implicated in gastrointestinal cancers, it would be valuable if the authors could elaborate on how iLBPs may interact with or influence ion channel–mediated oncogenic pathways.

We elaborate on how iLBPs may interact with ion channel–mediated oncogenic pathways in gastrointestinal cancers.

Finally, considering the role of ion channels in GI cancers, it is important to consider how iLBPs may interact with them. Ion channels are well known to be modulated by their lipid environment through both non-specific effects (membrane curvature, tension, hydrophobic mismatch) and specific lipid–protein interactions with cholesterol, polyunsaturated fatty acids, and phosphoinositides [142]. iLBPs are well positioned to influence channel activity by controlling the intracellular trafficking and availability of these lipid ligands. Consistent with this, in neuronal models FABP5 deletion increases the intracellular levels of endocannabinoids/NAE, thereby enhancing signalling through CB1 and PPARα receptors and ultimately suppressing NGF-driven TRPV1 sensitization [143]. Similarly, a FABP3 ligand (MF1) can potentiate GABAA currents by acting at the benzodiazepine recognition site leading to anti-seizure effects in vivo. These effects can be reversed by flumazenil confirming that MF1 acts as an allosteric modulator of the receptor [144]. In addition, FABP7 has been shown to influence glutamatergic ion channels. When FABP7 is deleted, docosahexaenoic acid (DHA) no longer enhances NMDA receptor activity in hippocampal neurons [145]. FABP7 deletion is also linked to greater NMDA receptor availability in striatal and cortical regions and mice lacking FABP7 show stronger locomotor responses when treated with the NMDA receptor blocker MK-801 [146, 147]. These studies show mechanisms by which iLBPs can directly or indirectly affect ion channel function. Although direct experimental evidence in GI cancers is currently lacking, these mechanistic studies suggest that iLBP-mediated lipid trafficking and ligand delivery could critically influence ion channel signalling, thereby opening a promising new direction for future research in GI cancer research.

Round 2

Reviewer 3 Report

Comments and Suggestions for Authors

It is well revised.